# RNA Interference Approach Is a Good Strategy against SARS-CoV-2

**DOI:** 10.3390/v15010100

**Published:** 2022-12-29

**Authors:** Ying-Ray Lee, Huey-Pin Tsai, Chun-Sheng Yeh, Chiung-Yao Fang, Michael W. Y. Chan, Tzu-Yun Wu, Cheng-Huang Shen

**Affiliations:** 1Department of Microbiology and Immunology, College of Medicine, Kaohsiung Medical University, Kaohsiung 807, Taiwan; 2Master of Science Program in Tropical Medicine, College of Medicine, Kaohsiung Medical University, Kaohsiung 807, Taiwan; 3Faculty of Post-Baccalaureate Medicine, College of Medicine, Kaohsiung Medical University, Kaohsiung 807, Taiwan; 4Center for Tropical Medicine and Infectious Disease, Kaohsiung Medical University, Kaohsiung 807, Taiwan; 5Department of Medical Laboratory Science and Biotechnology, College of Medicine, National Cheng Kung University, Tainan 701, Taiwan; 6Department of Pathology, National Cheng Kung University Hospital, College of Medicine, National Cheng Kung University, Tainan 701, Taiwan; 7Department of Medical Research, Ditmanson Medical Foundation, Chiayi Christian Hospital, Chiayi 600, Taiwan; 8Department of Biomedical Sciences, National Chung Cheng University, Min-Hsiung, Chiayi 621, Taiwan; 9Department of Urology, Ditmanson Medical Foundation Chiayi Christian Hospital, Chiayi 600, Taiwan

**Keywords:** SARS-CoV-2, COVID-19, shRNA, antiviral strategy

## Abstract

COVID-19, caused by SARS-CoV-2, created a devastating outbreak worldwide and consequently became a global health concern. However, no verifiable, specifically targeted treatment has been devised for COVID-19. Several emerging vaccines have been used, but protection has not been satisfactory. The complex genetic composition and high mutation frequency of SARS-CoV-2 have caused an uncertain vaccine response. Small interfering RNA (siRNA)-based therapy is an efficient strategy to control various infectious diseases employing post-transcriptional gene silencing through the silencing of target complementary mRNA. Here, we designed two highly effective shRNAs targeting the conserved region of RNA-dependent RNA polymerase (RdRP) and spike proteins capable of significant SARS-CoV-2 replication suppression. The efficacy of this approach suggested that the rapid development of an shRNA-based therapeutic strategy might prove to be highly effective in treating COVID-19. However, it needs further clinical trials.

## 1. Introduction

Since December 2019, severe acute respiratory syndrome coronavirus 2 (SARS-CoV-2) has caused the rapid global spread of the COVID-19 pandemic [1]. The most common clinical symptoms of SARS-CoV-2 infection include fever, dry cough, headaches, weakness, and shortness of breath. The pulmonary system is most prone to damage after SARS-CoV-2 infection, which may have a destructive effect on the lung tissue, causing severe acute respiratory disease and loss of life. SARS-CoV-2 can spread via respiratory droplets. Because infected individuals who are asymptomatic or exhibit mild symptoms can unknowingly spread the virus, the prevention of viral transmission is a significant problem [1]. So far, SARS-CoV-2 has caused more than 600 million infections and 6.5 million deaths. The demand for effective therapeutic drugs and preventive vaccines to avoid mortality and medical support overload has become very urgent [2]. Since December 2020, the Food and Drug Administration has, therefore, promulgated emergency approval for the use of some vaccines and antiviral drugs. However, the immunity against SARS-CoV-2 acquired after immunization is short-lived and limited. In addition, drugs against SARS-CoV-2 have limitations in specific COVID-19 treatments. Hence, the development of new SARS-CoV-2 treatment strategies is urgently needed [3,4].

SARS-CoV-2 is a positive single-stranded RNA virus belonging to the beta-coronavirus genus that includes SARS-CoV and MERS-CoV [5,6]. During infection, the binding of the SARS-CoV spike protein (SP) to the host cellular receptor angiotensin-converting enzyme 2 (ACE2) has been well demonstrated. After binding, the SP is cleaved by transmembrane serine protease 2 (TMPRSS2) into S1 and S2 subunits, which leads to viral and cellular membrane fusion to facilitate viral entry. Once the virus enters the cell, the viral RNA is released into the host cells. The positive-sense single-stranded RNA acts as a messenger RNA and hijacks the host machinery to undergo translation and viral replication.

RNA interference (RNAi) was first described in 1998, and most eukaryotic cells can use the RNAi pathway to target foreign double-stranded RNA (dsRNA) [7]. Defense is accomplished via the dsRNA region processing of pathogenic RNA into small or short interfering RNAs (siRNAs). In many organisms, RNAi is a major defense mechanism against pathogens. The RNAi approach is known to provide a tractable and specific target for antiviral treatment in numerous human viruses, including hepatitis B and C viruses, human immunodeficiency virus type 1, influenza viruses, poliovirus, and dengue virus [8]. siRNA also exerts effective antiviral activity, and since 2003, SARS-associated coronavirus replication inhibition has been reported [9]. Various recent studies have documented siRNA treatment for SARS-CoV and MERS-CoV, as well as SARS-CoV-2 in vitro and/or in vivo [3,10,11]. Notably, Li et al. demonstrated that using siRNA to target both RNA-dependent RNA polymerase (RdRp) and the SARS-CoV SP exerts the best prophylactic and therapeutic effect in Rhesus macaque SARS models [12]. Therefore, RNA interference used as a therapeutic strategy against COVID-19 seems straightforward and useful. However, its use remains limited and is the target of active research.

Frequent SARS-CoV-2 mutation is a troubling issue that compromises the vaccine’s immune protection and is a major concern in developing antiviral therapeutic agents. Because of the benefits of siRNA—including convenient design and production—it exerts more power than the existing vaccines and antiviral compounds developed to target infectious diseases, including COVID-19. Its design specifically targets SARS-CoV-2 according to the specific viral sequence, which appears to vary based on viral mutation.

In the present study, based on the SARS-CoV-2 (Wuhan-Hu-1; accession No. NC-045512) sequence, we designed two shRNAs targeting the SP and RdRp and transiently transfected them into Vero-E6 cells. We examined the suppression of SARS-CoV-2 viral replication in the cells under shRNA transfection using real-time polymerase chain reaction (PCR) and plaque assay. We demonstrated that shRNA reduced the viral replication under shRNA-specific targeting.

## 2. Materials and Methods

### 2.1. Viruses, Cell Line, and Cell Culture

We obtained clinical isolates of SARS-CoV-2 (hCoV-19/Taiwan/NCKUH-002/2020, full-length genome sequence GISAID No: EPI_ISL_493198) from a patient with COVID-19 from National Cheng Kung University Hospital and stored them at −80 °C. We purchased Vero-E6 cells from the American Type Culture Collection (Manassas, VA, USA) and cultured them in high-glucose-containing Dulbecco’s Modified Eagle Medium (Gibco, Grand Island, NY, USA) supplemented with 10% fetal bovine serum (Sigma, St. Louis, MO, USA), 100 U/mL penicillin (Sigma), and 100 µg/mL streptomycin (Sigma), as previously described.

### 2.2. Ethics Statement

The Institutional Review Board of National Cheng Kung University Hospital (No. B-ER-111-050) approved this study. The clinical and demographic information of the patient was unlinked before analysis, and informed consent was waived.

### 2.3. shRNA Construction

We designed plasmids (pSIREN-shCoV2019SP and pSIREN-shCoV2019RdRP) containing shRNAs targeting the SARS-CoV-2 (Wuhan-Hu-1 strain; accession No. NC-045512) SP and RdRp in May 2020, and Biotools Inc. (Taipei, Taiwan) produced them. The targeting sequence was predicted using siDirect, and the sequences were as follows (the targeting sequence was shown with underline):

pSIREN-shCoV2019SP:

5′-CACCGCACACGCCTATTAATTTAGTCTCGAGACTAAATTAATAGGCGTGTGC-3′

pSIREN-shCoV2019RdRP:

5′-CACCGCTAAACATGACTTCTTTAAGCTCGAGCTTAAAGAAGTCATGTTTAGC-3′

### 2.4. Cellular Viability Assay

We seeded Vero-E6 cells (5 × 10^5^) in a 6 cm dish overnight and transfected them with pSIREN-shCoV2019SP, pSIREN-shCoV2019RdRP, or empty plasmid (pBSSK^+^) with Lipofectamine 3000 (20 μL/dish; Thermo Fisher Scientific; Waltham, MA, USA). We assessed the cell viability with the CCK-8 assay kit, as in our previous report, 24 and 48 h post-transfection [13,14].

### 2.5. Detection of Viral Load with Real-Time PCR

We seeded Vero-E6 cells (1.6 × 10^6^) in a 25T flask (TPP Tissue Culture Flask) overnight and transfected them with pSIREN-shCoV2019SP, pSIREN-shCoV2019RdRP, or empty plasmid (pBSSK^+^) using a transfection reagent (Lipofectamine 3000; 20 μL/flask). At 24 h post-transfection, we infected the cells with SARS-CoV-2 at a multiplicity of infection (MOI) of 0.1. We determined the viral titer in the culture supernatant with real-time PCR. We used the viral RNA mini kit (QIAGEN, Hilden, Germany) to extract the viral RNA in the culture supernatant. We followed the real-time PCR procedure reported previously [15]. The real-time PCR method was used for relative quantification in this study. The known concentration of standard nucleic acid (or plasmid) to set up the standard curve. Using the standard curve to calculate the concentration of the unknown sample.

We cloned the target regions of plasmids are the spike protein gene and RdRP gene. The sequence of primers and probes of real-time PCR were listed as follows:

RdRP Forward:

5′-GTGARATGGTCATGTGTGGCGC-3′

RdRP Reverse:

5′-CAAATGTTAAAAACACTATTAGCATA-3′

RdRP probe:

5′-CAGGTGGAACCTCATCAGGAGATGC-3′

E Forward:

5′-ACAGGTACGTTAATAGTTAATAGCGT-3′

E Reverse:

5′-ATATTGCAGCAGTACGCACACA-3′

E probe:

5′-ACACTAGCCATCCTTACTGCGCTTCG-3′

### 2.6. Detection of Viral Load with Plaque Assay

We seeded Vero-E6 cells (1.6 × 10^6^) in a 25T flask overnight and transfected them with pSIREN-shCoV2019SP, pSIREN-shCoV2019RdRP, or empty plasmid (pBSSK^+^) using a transfection reagent (Lipofectamine 3000; 20 μL/flask). At 24 h post-transfection, we infected them with SARS-CoV-2 at an MOI of 0.1. We determined the viral titer in the culture supernatant with plaque assay using Vero-E6 cells. We followed the plaque assay procedure described previously [16].

### 2.7. Statistical Analysis

We presented the results as mean ± S.E.M. of at least three independent measurements. We used the two-way ANOVA analysis of variance and Fisher’s least significant difference test to analyze statistical significance. We considered a *p*-value of <0.05 to be statistically significant.

## 3. Results

### 3.1. ShRNA against SARS-CoV-2 Was Designed to Target Conserved Regions of SP and RdRp Genes

The RNAi approach against SARS-CoV, MERS-CoV, and SARS-CoV-2 to suppress viral replication has been demonstrated in various reports [3,10,11,12]. Moreover, Li et al. demonstrated that targeting the RdRp and SP of SARS-CoV exerts the best prophylactic and therapeutic effect in a Rhesus macaque SARS model [12]. Based on this finding, we evaluated the application of the shRNAs against SARS-CoV-2 in vitro. We compared the conserved sequences of SARS-CoV-2 Wuhan-Hu-1 strain (accession No. NC-045512) with SARS-CoV using BLAST and used them as the targeting sequence for shRNA construction. Finally, we designed the shRNAs against the RdRp and SP using siDirect (Figure 1).

Because of the high mutation frequency of SARS-CoV-2, we evaluated the efficacy of our shRNA designed in May 2020 in targeting the conserved regions of the SP and RdRp genes in various SARS-CoV-2 strains. The National Center for Biotechnology Information reported the complete genome sequence database of SARS-CoV-2 as numbering over 1.6 million. Instead, we used the representative SARS-CoV-2 strains (29 representative strains with the Wuhan-Hu-1 strain as the reference) in the Bacterial and Viral Bioinformatics Resource Center (BV-BRC) to compare using nucleotide BLAST. Figure 1 identifies conserved regions of the designed shRNA via BLAST. The data demonstrated that the targets of the designed shRNAs in this study were highly conserved in various SARS-CoV-2 strains. Of these, pSIREN-shCoV2019RdRP showed 100% targeting on all SARS-CoV-2 strains, and pSIREN-shCoV2019SP showed targeting on all SARS-CoV-2 strains, except the Omicron strain (Figure 1).

### 3.2. ShRNA against SARS-CoV-2 Is Nontoxic in Vero-E6 Cells

To evaluate the inhibitory effects of shRNA constructions in SARS-CoV-2, we investigated the plasmid transfection efficiency with a reporter plasmid, pEGFPC1. We transfected the plasmid with lipofectamine 3000 at various dosages into Vero-E6 cells and used the green fluorescent protein (GFP) expression as an indicator of transfection efficiency after determining it under fluorescence microscopy (Olympus; Tokyo, Japan). Figure 2 shows that the transfection efficiency increased in a dosage- and time-dependent manner in Vero-E6 cells.

Furthermore, we performed a cellular viability analysis to address the safety of the shRNA expression in Vero-E6 cells. Here, we demonstrated that transfection and SARS-CoV-2 shRNA overexpression showed nontoxicity in Vero-E6 cells (Figure 3). This data suggests that the SARS-CoV-2 shRNAs are safe for Vero-E6 cells.

### 3.3. ShRNAs Targeting SARS-CoV-2 Exert Antiviral Replication Activity in Vero-E6 Cells

To address the shRNA application against SARS-CoV-2 in a clinical setting, we transfected the pSIREN-shCoV2019SP and pSIREN-shCoV2019RdRP constructions into Vero-E6 cells and determined the viral replication in the cells using real-time PCR or the viral titer using plaque assay in the culture supernatants after infection with a clinically isolated SARS-CoV-2 virus (not yet sequenced). The data showed significant suppression of viral replication in the cells with pSIREN-pSIREN-shCoV2019SP and pSIREN-shCoV2019RdRP transfections (Figure 4A), which suggests that the shRNA of the SP and RdRp exerted a strong inhibitory effect on the clinically isolated unknown SARS-CoV-2 strain. Furthermore, the shRNA of the SP and RdRp could significantly inhibit the viral titer in the culture supernatant (Figure 4B), which showed that the shRNAs of both the SP and RdRp in this study can effectively suppress SARS-CoV-2 replication. To understand how shRNAs of the SP and RdRp in this study effectively targeted this clinically isolated unknown virus, we determined the complete sequence of the virus, and BLAST showed that shRNAs of both the SP and RdRp could target this virus (hCoV-19/Taiwan/NCKUH-002/2020) as well as the Wuhan-Hu-1 strain, with 100% efficiency (Figure 5).

These data illustrate that shRNA targeting SARS-CoV-2 suppresses viral replication if the virus-targeting sequence is expressed. Our study demonstrated that designing an shRNA target on a conserved sequence can be a useful preventive or therapeutic strategy for anti-SARS-CoV-2 infection. Moreover, targeting the conserved region of SARS-CoV-2 RdRp is better than targeting that of the SP because the mutation of the SP sequence of the Omicron strain can escape the shRNA targeting even if it can be targeted on most SARS-CoV-2 strains.

## 4. Discussion

The COVID-19 pandemic has aggressively spread globally from 2019 to the present. Diverse treatment approaches, including vaccines, drugs, antagonist peptides, and monoclonal antibodies, can be used as prophylactic or therapeutic strategies to confront this pandemic. Paxlovid and molnupiravir, are available for outpatients with mild to moderate COVID-19 to against hospitalization and death; they must begin within five days of symptom onset to maintain product efficacy [17]. In addition, monoclonal antibody-based COVID-19 treatments, including bamlanivimab, etesevimab, casirivimab, imdevimab, sotrovimab, and bebtelovimab; only bebtelovimab has been reported to retain its efficacy against all SARS-CoV-2 variants [18]. However, the high mutation frequency of the virus allows it to evade these preventive and therapeutic strategies [19]. Therefore, strategies that can adapt in response to the high mutation frequency of SARS-CoV-2 are urgently needed.

The high mutation frequency of SARS-CoV-2 poses a serious challenge to antiviral agents or antagonist peptide development. Compared with classical vaccines, mRNA vaccines have the advantage of rapid preparation in response to viral antigenic mutations, but their effectiveness is still limited [20]. In the mRNA vaccine’s design, the SP has been used as the antigen for vaccine development. However, many mutations in the SP, including mutations in the receptor-binding domain (RBD) or the SP N-terminal domain in the Omicron variant [21], which cause evasion of immunity and reduction in vaccine effectiveness, have been determined [22,23]. The waning immunity that occurs over time is another issue. The rapid decline of the neutralizing antibodies in the first 3 months after boosting with mRNA vaccines causes waning immunity and reduces the vaccine’s effectiveness [24,25]. Therefore, the imminent development of an alternative preventive or therapeutic strategy that can respond to the high mutation frequency of SARS-CoV-2 is urgently required.

RNAi, an excellent tool, has been proven to silence gene expression in various species, causing post-transcriptional gene silencing by binding to the complementary mRNA sequence and leading to its enzymatic degradation [26,27]. SiRNA, one of the RNAi approaches, can be generated with shRNA, a double-stranded ncRNA molecule comprising 20–25 bp, which can reduce the expression of targeted genes. Moreover, siRNA-based therapies have been developed and implemented for anticancer, antiviral, and antibacterial infections and genetic disease purposes [26]. Recent studies, including this study, have demonstrated that the siRNA approach can be a highly effective strategy against SASR-CoV-2 infection and reduces viral loads via replication suppression [28,29,30]. Although siRNA targeting RdRp and the SP is predicted to be a good candidate against SARS-CoV-2, these prediction studies are limited because they have no evidence for viral infection studies [28,29].

On the contrary, COVID-19 is an emerging infectious disease with a high mutation frequency. Therefore, the siRNA approach can be used to respond to the mutations using the conserved sequences in the design approach. As the viral strain in patients is unknown, the siRNAs clinically used against SARS-CoV-2 have to target various SARS-CoV-2 strains effectively. More importantly, the designed siRNA can be used against SARS-CoV-2 infections clinically. Designed siRNAs were transfected into cells and infected with a clinically isolated unknown viral strain. This study’s significance is that shRNA targeting was demonstrated on both RdRp and SP and could considerably reduce viral replication in the infected cells and the titers of viral progenies in the culture supernatant (Figure 4). In addition, we evaluated the nontoxicity of these shRNAs in the transfected cells, and our results suggested that no off-targeting toxicity occurred. These data suggest that the designed shRNAs have immense potential to be used clinically for SARS-CoV-2 infection prophylaxis or treatment.

Because the shRNAs used in this study were designed in May 2020 and the Omicron strain appeared in Nov 2021, the retrospective comparison in Figure 1 showed that the mutation of the SP conserved sequence can also evade shRNA targeting. Therefore, we suggest using the shRNA cocktail to provide better protection or therapeutic effects for COVID-19 [31]. Four siRNAs have been demonstrated to bind complementary sequences in the SARS-CoV-2 genome and presented in every clinically relevant published SARS-CoV-2 genome, including variants α, β, γ, and Omicron, effectively reducing viral load [32]. Moreover, a siRNA formulation with an available delivery/adjuvant—such as intranasal administration in clinical settings—may serve as a powerful prophylactic or therapeutic application.

## Figures and Tables

**Figure 1 viruses-15-00100-f001:**
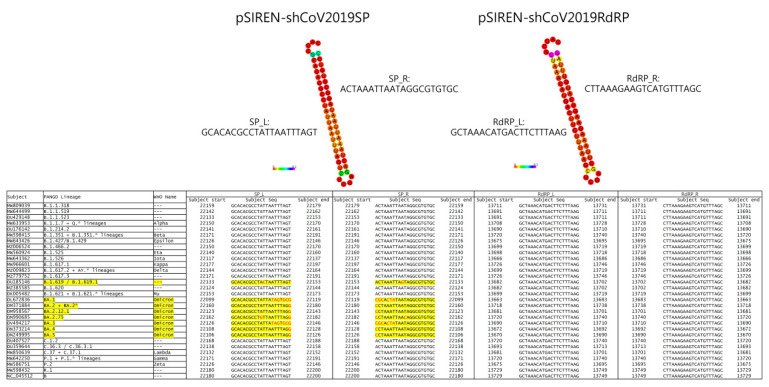
Conservation of targeting of shCoV2019SP and shCoV2019RdRP among different variants of SARS-CoV-2.

**Figure 2 viruses-15-00100-f002:**
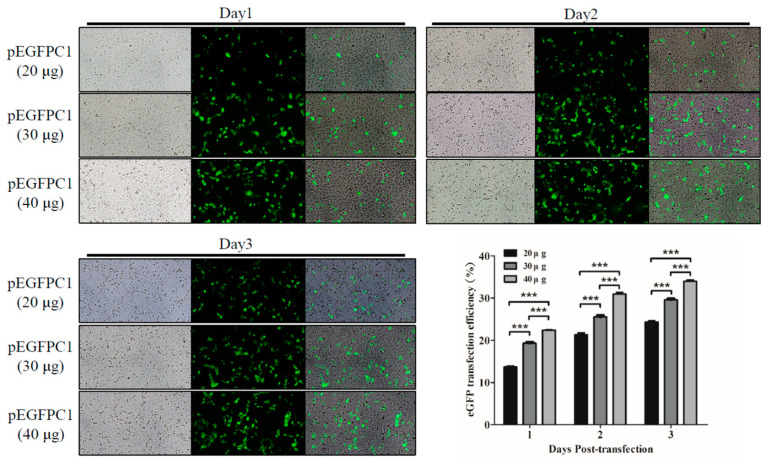
The transfection efficiency in Vero-E6 cells. We transfected the cells with pEGFPC1 at the indicated dosages using lipofectamine 3000. After transfection, we detected the transfection efficiency with GFP expression under fluorescent microscopy. Three independent experiments were performed. *** *p* < 0.001.

**Figure 3 viruses-15-00100-f003:**
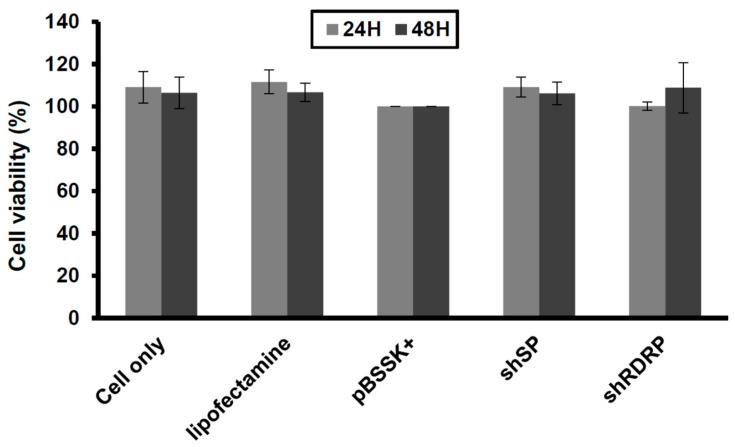
Overexpression of shRNA is safe in Vero-E6 cells. We transfected the cells with the shRNA constructions (pSIREN-shCoV2019SP and pSIREN-shCoV2019RdRP; 40 μg) with lipofectamine 3000 and assessed the cellular viability with a CCK-8 assay kit. We used lipofectamine and pBSSK^+^ construction (40 μg) as a negative control and standard. Three independent experiments were performed.

**Figure 4 viruses-15-00100-f004:**
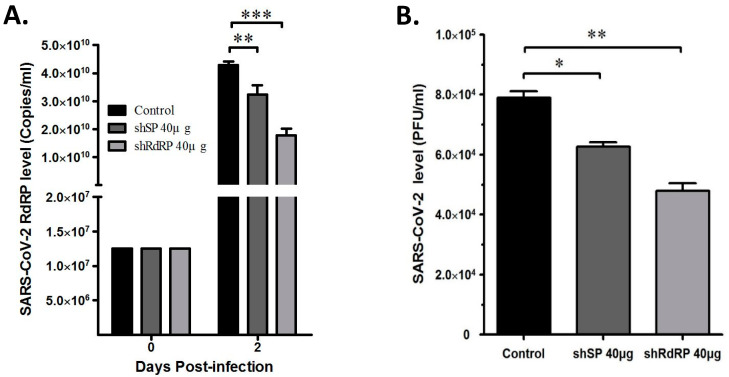
shRNAs targeting a clinically isolated unknown SARS-CoV-2 virus inhibited viral replication. We transfected Vero-E6 cells with pBSSK^+^, pSIREN-shCoV2019SP, or pSIREN-shCoV2019RdRP and infected the cells with a clinically isolated unknown SARS-CoV-2 strain. We assessed the viral replication and viral titer in the culture supernatant with (**A**) real-time PCR and (**B**) plaque assay 2 days post-infection. We used pBSSK^+^ as a negative control. Three independent experiments were performed. * *p* < 0.05. ** *p* < 0.01. *** *p* < 0.001.

**Figure 5 viruses-15-00100-f005:**
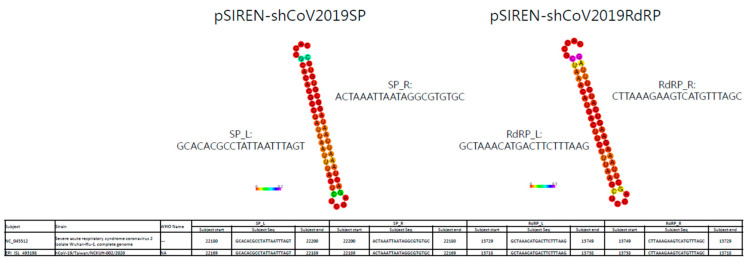
Gene sequence alignment of RdRP and SP shRNA vs. SARS-CoV-2 viruses. We compared the sequence alignment of viral sequences, including the Wuhan-Hu-1 strain and the hCoV-19/Taiwan/NCKUH-002/2020 strain, with the RdRP and SP shRNA targeting sequences.

## Data Availability

Not applicable.

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
