# Peer review of "RNA Interference Approach Is a Good Strategy against SARS-CoV-2"

_viruses, 2022, doi:10.3390/v15010100_

Round 1
Reviewer 1 Report
Authors have reported RNA interference approach is a good strategy for against SARS-CoV-2 article is good but some points needs to be considered
1. Authors may discuss in the introduction recent RNA interference in Omicron they can refer the following articles https://doi.org/10.3390/ijms23052408 ; https://doi.org/10.1002%2Fjmv.27936; https://www.frontiersin.org/articles/10.3389/fbioe.2020.00916/full
2. Has the authors compare the results with any standard drug/compounds or any other vaccines.
3. This approaches is effective against all the type of covid virus
4. Conclusion and future direction should be given.
5. Spelling and grammatical errors need to be checked.
6. Authors instruction must be followed.
Reviewer 2 Report
- Line 31-32: It is necessary to rewrite this item, as it is not possible to guarantee full effectiveness using only in vitro tests.
- Line 50: Describe treatments used such as Paxlovid and therapeutic antibodies that are effective routine treatments
- Line 132: How much lipofectamine was used? Was each target used individually, or was a pool performed? It is necessary to clarify the information about the administration of shRNA and lipofectamine.
- Line 139: Was ANOVA performed without a post-test? Describe which groups were compared.
- Line 113: The use of a figure with the groups tested and when they were collected can facilitate the understanding of the groups used.
- Line 128: Was an absolute quantification used? Which regions were cloned to perform the standard curve?
- Figure 3: The cell viability shown demonstrates that even where there is only the study cell, the value in percentage is greater than 100%, how do you explain this?
- Figure 04: In your method you describe that the viral load was performed by RT-qPCR, however in Figure 4-B you use the PFU/ml unit that is normally used for plating cells. Can you better describe this information.
- Line 267: Other studies using siRNA cocktail have already been tested and used (https://www.mdpi.com/2073-4425/13/11/2147/html) I believe this work can help you to emphasize the need for siRNA cocktail for use in SARS-CoV-2. In addition, the author also outlines interesting viral breakout points to address in his discussion.
Round 2
Reviewer 1 Report
Comments incorporated
Reviewer 2 Report
Dear,
The manuscript has greatly improved its arguments and become more tenable in its claims.
However, the following adjustments are necessary:
1 - The sequences of sh must be inserted in tables. As well as inserting the location of the genome that it binds
2 - insert the PCR primers into a table
3 - in Statistical analysis put the programs used
4 - In figure 3 there is a symbol on the Y axis, please remove
5 - Line 346 has a quote error.
